# Intelligent Learning-Based Methods for Determining the Ideal Team Size in Agile Practices

**DOI:** 10.3390/biomimetics9050292

**Published:** 2024-05-13

**Authors:** Rodrigo Olivares, Rene Noel, Sebastián M. Guzmán, Diego Miranda, Roberto Munoz

**Affiliations:** Escuela de Ingeniería Informática, Universidad de Valparaíso, Valparaíso 2362905, Chile; rene.noel@uv.cl (R.N.); sebastian.medina@postgrado.uv.cl (S.M.G.); diego.mirandah@postgrado.uv.cl (D.M.); roberto.munoz@uv.cl (R.M.)

**Keywords:** metaheuristics, machine learning, ensemble learning, agile practices, software engineering

## Abstract

One of the significant challenges in scaling agile software development is organizing software development teams to ensure effective communication among members while equipping them with the capabilities to deliver business value independently. A formal approach to address this challenge involves modeling it as an optimization problem: given a professional staff, how can they be organized to optimize the number of communication channels, considering both intra-team and inter-team channels? In this article, we propose applying a set of bio-inspired algorithms to solve this problem. We introduce an enhancement that incorporates ensemble learning into the resolution process to achieve nearly optimal results. Ensemble learning integrates multiple machine-learning strategies with diverse characteristics to boost optimizer performance. Furthermore, the studied metaheuristics offer an excellent opportunity to explore their linear convergence, contingent on the exploration and exploitation phases. The results produce more precise definitions for team sizes, aligning with industry standards. Our approach demonstrates superior performance compared to the traditional versions of these algorithms.

## 1. Introduction

The agile approach to software engineering is based on a series of practices that prioritize face-to-face collaboration between members of the software development team [1]. While this allows one to address the change, considering it as the only constant in an uncertain environment [2], it poses a challenge in terms of scalability as the complexity of communication increases with a larger number of team members.

The recent studies in high-performance technological organizations have shown that small, self-organized, and independent software teams are more efficient in software delivery [3]. On the other hand, frameworks for scaling agile development practices promote the formation of independent self-organized teams to enable scalability of agile development [4,5]. This modularization translates into a modularization of the system’s architecture, where approaches like domain-driven design [6] and microservices [7] advocate that small teams handle a reduced part of the problem domain to maintain technical simplicity and reduce the cognitive load of the development team. The impact of communication within and between teams on software development efficiency is significant, to the extent that digital transformation frameworks [8,9] encourage designing the structure of software teams as if it were the architecture of software components: the less coupled the communication of the teams, the less coupled the software components are.

In this context, a relevant question arises: What is the ideal size of a software development team? Although this question has been approached with anecdotal approximations like the two-pizza rule (a team should not be larger than the number of people two pizzas can feed) [10], there are various factors in the interactions among team members that can influence the team’s size. A systematic approach to this problem is proposed in [11]. In this study, the challenge is addressed as an optimization problem: given the number of professionals, how to organize them to optimize the number of communication channels, considering both the channels within the team and between teams. For that, a conventional genetic algorithm (GA) is employed. The results obtained after the GA optimization process indicate that the optimal distribution of teams for the total project members ranges from five to nine members per team. However, there is room for improvement in the aforementioned results by employing new artificial intelligence techniques. These techniques enable the use of historical data to achieve better outcomes when determining the distribution of team sizes in an agile practices project.

In artificial intelligence, metaheuristic optimization algorithms have emerged as powerful techniques capable of solving complex optimization problems with reduced solving times [12]. Bio-solvers go through two main phases, known as exploration and exploitation. Many of these algorithms have been designed to achieve better results through these two stages. However, when this is not the case, the mechanism within the algorithm may have some slight disadvantages in balancing exploration and exploitation, leading to suboptimal performance [13]. For this reason, several works have been proposed in this line of research to balance exploration and exploitation better and achieve more efficient solutions.

Techniques based on learning mechanisms to support optimization metaheuristics have been successfully applied to problems involving the trade-off between exploration and exploitation [14]. However, in many cases, the procedure guiding nature-inspired algorithms operates individually; that is, it is a single technique that enhances the optimization algorithm. Our work proposes a new balance between exploration and exploitation using ensemble learning (EL). This strategy involves integrating a machine-learning technique with specific characteristics to achieve superior results. As surrogate optimization algorithms, we employ seven well-known bio-inspired techniques: particle swarm optimization, gray wolf optimizer, firefly algorithm, bat algorithm, whale optimization algorithm, crow search algorithm, and differential evolution. We chose these mechanisms because they exhibit a linear characteristic of avoiding solutions trapped in local optima while balancing the exploration and exploitation phases.

Ensemble learning into the optimization algorithm reveals that the optimal team size and distribution for a given total number of team members range between six and eight members per team, which aligns with industry practices [15,16,17]. These results are practically equal to those reported in [11].

The rest of the article is organized as follows: Section 2 presents the state of the art, including relevant studies. Section 3 covers important concepts considered in the proposal. Next, the developed solution is presented in Section 4. Section 5 defines the computational experimental setup. The results and discussion are exposed in Section 6. Finally, Section 7 provides a critical outlook on the results achieved and outlines the potential future work.

## 2. Related Work

Traditional project management approaches often try to achieve the objectives of innovative projects. It is challenging to incorporate detailed initial plans in projects that produce innovations, such as new products and software, due to the complexity of requirements and solution planning and the lack of information from previous similar projects [18]. The information scarcity in such projects makes them uncertain for traditional management methods [19,20]. Agile project management addresses these shortcomings. It involves adaptive team skills regarding scope and solution requirements, distinct leadership involvement, and the ability of agile teams to be creative and innovative [21]. However, this heavily relies on active commitment from team members to the project tasks [22].

Few studies have focused on improving communication and collaboration within agile teams. One of them is published in [11]. The proposal solves an optimization model for agile team sizes using the genetic algorithm. The objective is to develop a GA model to find the optimal size of a self-organized team, considering team communication characteristics. The model uses the number of communication channels between individuals and teams as the objective function. Another work, [23], proposes an optimization model for agile team relationships based on role patterns and fuzzy logic. Role patterns evaluate team cohesion from three perspectives: the team competence complement index, role typology matching index, and team typology matching index. The model aims to determine the most suitable candidate for a vacant position and effective team cohesion. Fuzzy logic is used for the linguistic evaluation of system output, classifying results as low, medium, or high. While this model has not been tested with metaheuristic algorithms, fuzzy logic offers the advantage of defining and measuring imprecise information about the candidates’ psychological and behavioral traits, which is difficult to model using conventional mathematics.

Concerning diversity methods, many strategies have been explored to achieve a balance between exploration and exploitation. In [24], a compilation and investigation of works related to exploration and exploitation in bio-inspired algorithms within the branch of evolutionary algorithms is presented. According to this study, the same concept will be applied to swarm intelligence, using different diversity measures to define neighborhood relationships, which are necessary to delineate exploration from exploitation. For this purpose, we focus on maintaining and controlling diversity through various techniques. These techniques are divided into two main branches within diversity maintenance: non-niching and niching. Any technique that maintains diversity in the population based on the distance between the population members is considered a niching technique. Non-niching and niching methods can maintain a population of diverse individuals, while niching methods can also locate multiple optimal solutions.

Several works analyze how metaheuristics are improved using machine learning, regression, and clustering techniques [25,26,27,28]. In [29], a machine learning model that predicts solution quality for a given instance is created using the support vector machine. After that, this approach modifies the parameters and directs the metaheuristics to more fruitful search areas. In [30], the authors provide an evolutionary algorithm that regulates operators and parameters. For this, a controller module that applies learning rules assesses the impact and assigns restarts to the parameter set is integrated. Similar to these, the research presented in [31] investigates how to combine the variable neighborhood search algorithm with reinforcement learning, employ reactive strategies for parameter change, and choose local searches to balance the exploration and exploitation stages.

Other works published in [32,33] combine PSO with regression models and clustering strategies for population management and parameter adjustment, respectively. Another classifier-based PSO is provided in [34] with the aim of an optimization approach that does not depend on its parameters. This method enhances the exploration of the search space and the quality of the solutions retrieved by classifying the solutions discovered by the particles using a previously trained model.

In line with other efforts, the PSO is once again improved with a learning model to regulate its parameters, yielding a competitive performance compared to alternative parameter adaption algorithms [35]. This approach is also applied in GWO [36]. Here, RNA crossover operations and an adaptive parameter control scheme were employed. Here, each individual wolf learns from its neighbors, becoming another candidate for its new position. In [37,38,39], randomized balancing strategies are applied to GWO. These improvements are integrated into the local search procedure in order to balance the exploration and exploitation phases.

Another bio-inspired algorithm enhanced by a learning mechanism is the firefly optimization. In [40], the authors propose a multi-strategy ensemble to effectively achieve a better balance between exploration and exploitation. Furthermore, in [41], an ML-based work is presented to guide new population generation in an evolutionary process. The proposed approach, called the learnable evolution model, executes ML to uncover reasons for the superior performance of specific individuals in designated tasks. These inductive hypotheses are then used to create a new solution generation.

From a similar perspective, some works in machine learning, such as [42], offer an overview of the fusion of data mining techniques and metaheuristics. The application of these methods within metaheuristic algorithms can be observed in various stages, including initializing solution(s), managing solution(s) during the search process, and integrating data mining into operators and local searches.

The reviewed literature underscores the diverse approaches and innovations in combining metaheuristics with machine learning techniques. These studies collectively highlight the potential for enhanced problem-solving, optimization, and adaptive strategies through the synergy of these two domains. The synthesis of these methodologies contributes to the theoretical understanding of optimization and presents practical avenues for addressing complex real-world challenges. The following sections delve into the proposed method and experimental evaluation, building upon the insights gained from these related works.

## 3. Preliminaries

This section first introduces the ideal time-size problem for agile practices. Next, we expose the ensemble learning paradigm that includes classifying, training, and combining models. Finally, we present the integration between the learning-based approach and different swarm intelligence methods.

### 3.1. Time-Size Problem

Agile practices are a mechanism to reduce costs and respond to changes in dynamic market conditions [43]. It involves continuous collaboration with stakeholders and ongoing improvement at each project stage. When starting a project, teams go through planning, execution, and evaluation. Continuous collaboration among team members and project stakeholders is vital, as it impacts communication and performance within the work [43]. Several frameworks have been developed, including Extreme Programming [44], SCRUM [45], Adaptive Project Framework [46], and Crystal Methodologies [47]. These agile principles are implemented to develop products and project teams that can swiftly adapt to changes, thereby increasing organizational agility and achieving significant business benefits in a fluctuating business environment. Notably, it points out that while empirically determining a suitable team size is useful, it is often insufficient for optimizing team efficiency, necessitating more dynamic and scientifically grounded approaches.

To address the limitations of empirical methods in determining team size, an optimization model is employed [11]. This model aims to minimize the total number of communication channels among team members, thereby reducing the complexity and potential for inefficiencies in team interactions. The goal is to find the optimal equipment configuration to reduce communication load, streamline processes, and enhance team efficiency.

Initially, the model establishes the basic parameters: *N* represents the total number of communication channels among all team members, *T* denotes the number of teams, and *n* specifies the number of members in each team, defined by the set $\left\{t_{1},t_{1},\ldots ,t_{n}\right\}$. After, for each team, the total number of communication channels $N_{t}$ is calculated using the formula $N_{t}=\left[n\left(n-1\right)/2\right]$. This equation calculates the combinations of team members that can communicate within the same team based on the assumption that every member communicates with every other member. Next, the total number of communication channels between different teams $N_{T}$ is computed as $N_{T}=\left[T\left(T-1\right)/2\right]$. This accounts for each team needing to communicate with every other team at least once. Finally, the comprehensive total *N* is then given by $N=T\times N_{t}+N_{T}$.

To reflect the diverse impact of various communication methods, the model incorporates weighted factors for different communication types: $w_{1}$ for person-to-person verbal communication, $w_{2}$ for team-to-team communication, $w_{3}$ for email communications, $w_{4}$ for video conference communications, and $w_{5}$ for multilingual communications.

The objective function to minimize the communication load is formulated as follows:(1)$$\begin{array}{c} min\;\sum_{i=1}^{T}\left(\frac{n_{i}\left(n_{i}-1\right)}{2}\right)\times p_{w}+\left(\frac{T(T-1)}{2}\right)\times t_{w}\\ s.t\\ \sum_{i=1}^{T}n_{i}=N \end{array}$$
where $p_{w}$ includes weights for personal communications $\left\{w_{1},w_{3},w_{5}\right\}$, and $t_{w}$ accounts for team-based communications $\left\{w_{2},w_{4}\right\}$. The weights $w_{i}$ are empirically sourced from the prior research and are set between 1 and 1.5, providing a realistic approximation of the communication impacts [11].

The optimization problem of determining the ideal team size within agile practices is combinatorial, as it aims to optimize communication channels within and between teams. Given the number of team compositions possible from a pool of project members, the search space grows exponentially with the increase in team and project size. This complexity is further compounded when considering constraints such as minimum and maximum team sizes. For instance, if a project involves *n* members and each team must have between *a* and *b* members, the total number of potential team configurations can be calculated using combinatorial methods. Therefore, The search space size is factorial, growing exponentially with the increase in *n*. This exponential growth exemplifies the NP-hard nature of the problem, as the number of potential configurations increases rapidly, even with a small increase in the number of team members.

This combinatorial explosion suggests that our team size optimization problem can be classified as NP-hard. The vast array of potential configurations presents significant theoretical and computational challenges, necessitating an efficient approach to exploring such a large search space.

In response to these challenges, we employ bio-inspired algorithms, particularly enhanced with ensemble learning. These algorithms are well-suited for efficiently searching large, complex spaces and offer a pragmatic approach to achieving near-optimal solutions within acceptable time frames and computational resources. The enhancements through ensemble learning help balance the exploration and exploitation phases of the algorithms, leading to more precise definitions of team sizes that align with industry standards. This strategic methodological choice aligns with the need to efficiently handle inherent complexity and scalability challenges while ensuring that the solutions are practical and implementable in real-world scenarios.

### 3.2. Ensemble Learning

Ensemble learning is a robust multiple classification system that leverages the strength of various learning models to create a superior composite model. As detailed in [48], ensemble methods synthesize the predictive power of several base learners to enhance predictive accuracy and model stability over individual classifiers.

The concept of homogeneous and heterogeneous ensembles delineates the landscape of ensemble learning. Homogeneous ensembles, such as decision tree ensembles or neural network ensembles, consist exclusively of one type of base learner, enhancing the predictive strength through diversity in model training procedures. Heterogeneous ensembles, in contrast, integrate diverse types of learners, broadening the scope of learning capabilities beyond what single-model ensembles can achieve.

The effectiveness of ensemble methods is captured in our adaptation of the basic EL concept illustrated in Figure 1, where multiple learners are trained, and their predictions are synergistically combined to produce superior results. This approach mitigates individual learner errors across varied datasets, as supported by [49], who emphasize the enhanced performance due to error decorrelation among learners.

In EL, two distinct approaches are utilized to facilitate the learning process for individual learners through training. One involves a sequential methodology, while the other employs a parallel system. To achieve this, two pivotal algorithms stand out, namely, *boosting* and *bagging*. Boosting employs a sequential methodology to elevate weak individual learners to a more robust level. On the other hand, bagging utilizes a parallel training method that emphasizes ensemble-based sampling. Our study extends these foundational techniques by exploring combinatorial methods within EL, aiming to harness and optimize the collective capabilities of diverse learning algorithms.

Combining individual learners at the end of the training process offers several benefits. These include mitigating the risk of selecting a single learner with mediocre results, avoiding poor generalization performance, enhancing outcomes at this stage, and enabling a more accurate approximation to the problem’s solution by amalgamating individual learners.

If an ensemble contains *L* individual learners $\left\{h_{1},h_{2},h_{3},\ldots ,h_{L}\right\}$, then the output of these is represented by $h_{i}\left(x\right)$, where *x* is the input sample [48]. The most commonly used methods for performing these combinations are as follows:Averaging: This approach is predominantly employed by individual learners engaged in regression tasks or those producing numerical outputs. It encompasses two distinct methods: simple averaging (see Equation (Equation 2)) and weighted averaging (see Equation (Equation 3)).
(2)$$H\left(x\right)=\frac{1}{T}\sum_{i=1}^{T}h_{i}\left(x\right)$$
(3)$$H\left(x\right)=\sum_{i=1}^{T}w_{i}h_{i}\left(x\right)$$In Equation (Equation 3), $w_{i}$ represents the weight of individual learner $h_{i}$, usually satisfying $w_{i}\geq 0$ and $\sum_{i=1}^{T}w_{i}=1$. Notably, simple averaging is a special case of weighted averaging when $w_{i}=\frac{1}{T}$. Weighted averaging proves more suitable when individual learners exhibit substantial performance disparities, while simple averaging is preferable when individual learners display similar performance.Voting: This strategy is primarily used for classification tasks where individual learners predict labels from a known set of labels using a voting approach. Various methods exist within this category, including majority voting (see Equation (Equation 4)), plurality voting (see Equation (Equation 5)), and weighted voting (see Equation (Equation 6)).
(4)$$H\left(x\right)=\left\{\begin{array}{cc} c_{j}, & \mathrm{if}\;\sum_{i=1}^{T}h_{i}^{j}\left(x\right)>0.5\\ \mathrm{reject}, & \mathrm{otherwise} \end{array}\right.$$
(5)$$H\left(x\right)=c_{\underset{j}{argmax}}\sum_{i=1}^{T}h_{i}^{j}\left(x\right)$$
(6)$$H\left(x\right)=c_{\underset{j}{argmax}}\sum_{i=1}^{T}w_{i}h_{i}^{j}\left(x\right)$$Combining Learning: This approach is employed when a substantial amount of data is available for training. It involves using a meta-learner to combine individual learners. A representative technique in this category is stacking. In stacking, individual learners are referred to as first-level learners, while the combined learners are known as second-level learners or meta-learners.The stacking process commences by training first-level learners using the original training dataset. A new dataset is then generated to train second-level learners. In this new dataset, outputs from first-level learners serve as input features, while the original training sets labels remain unchanged. To mitigate the risk of overfitting, cross-validation or leave-one-out methods are often employed during the generation of the second-level training set, ensuring that samples unused in training first-level learners contribute to the second-level learner’s training set.In this work, we employ the voting scheme because it combines machine learning classifiers and it averages predicted probabilities to forecast feature labels, enabling the use of varied models for enhanced classification results.

### 3.3. Bio-Inspired Algorithms

Metaheuristics are optimization techniques that draw inspiration from natural processes to solve complex problems in various domains [12]. These algorithms provide powerful tools for tackling problems where traditional optimization methods may fail due to their nonlinearity, high dimensionality, or multimodality. Among the plethora of metaheuristics, several stand out for their unique approaches and successful applications [50].

In our study, we selected bio-inspired algorithms based on their proven ability to effectively manage complex, multidimensional search spaces essential for optimizing agile team sizes. We utilized both cutting-edge and newly developed algorithms to ensure robustness and innovation in our approach. Key considerations included their rapid convergence rates and computational efficiency, vital for the fast-paced nature of agile projects. Additionally, these algorithms’ behaviors, which mimic the collaborative dynamics of agile teams, greatly improved their relevance and effectiveness for our research.

Finally, it is important to emphasize that all bio-inspired algorithms have been adapted for an integer domain. Originally designed to address continuous problems, these algorithms now incorporate a discretization method using the sigmoid function $\left[1/\left(1+e^{-x_{i}^{j}}\right)\right]$ to make them suitable for integer problems [51]. This process involves rounding each element of the solution vector $x_{i}^{j}$ to the nearest integer within a predefined range. Depending on the value derived from the sigmoid function, $x_{i}^{j}$ is assigned the closest integer. Through this transformation, we ensure that the variables of the solution vector strictly adhere to the problem’s integer constraints, maintaining them within the defined lower and upper bounds.

#### 3.3.1. Particle Swarm Optimization

Particle swarm optimization is a bio-inspired metaheuristic derived from the social behavior of birds flocking or fish schooling [52]. This algorithm models potential solutions as particles that navigate through the solution space by adjusting their positions and velocities based on personal and collective experiences. Each particle in the swarm represents a potential solution to the optimization problem and moves through the solution space guided by its own experience as well as the collective experience of the swarm.

Each particle updates its position by following two best values: the best solution it has achieved so far (personal best) and the best solution any particle in the population has achieved (global best). The movement of each particle is influenced by its velocity, which dynamically adjusts at each iteration of the algorithm according to the following equations:(7)$$v_{i}^{\left(t+1\right)}=w\cdot v_{i}^{\left(t\right)}+c_{1}\cdot r_{1}\cdot \left(p_{best,i}-x_{i}^{\left(t\right)}\right)+c_{2}\cdot r_{2}\cdot \left(g_{best}-x_{i}^{\left(t\right)}\right)$$
(8)$$x_{i}^{\left(t+1\right)}=x_{i}^{\left(t\right)}+v_{i}^{\left(t+1\right)}$$

In these equations, $v_{i}$ denotes the velocity of particle *i*, $x_{i}$ denotes its position, $p_{best,i}$ is the personal best position it has discovered, and $g_{best}$ is the best position found by any particle in the swarm. The parameters *w*, $c_{1}$, and $c_{2}$ represent the inertia weight and the cognitive and social scaling coefficients, respectively. These coefficients control the impact of the past velocities, the cognitive component (individual memory), and the social component (swarm influence), on the velocity update. The parameters $r_{1}$ and $r_{2}$ are random numbers generated anew for each update and for each particle, providing stochasticity to the search process and helping to escape local optima.

The inertia weight *w* plays a critical role in balancing the exploration and exploitation abilities of the swarm. A higher inertia weight facilitates exploration by encouraging the particles to roam further in the search space, while a lower inertia weight aids exploitation by allowing finer adjustments in the neighborhood of the current best solutions.

PSO has been successfully applied to a wide range of problems, from classical optimization problems such as function minimization and resource allocation to complex real-world applications like neural network training, electric circuit design, and multi-objective optimization. Its popularity stems from its simplicity, ease of implementation, and robust performance across diverse problem domains. Additionally, PSO’s mechanism is inherently parallel, making it suitable for implementation on parallel and distributed computing architectures, further enhancing its efficiency and scalability.

#### 3.3.2. Gray Wolf Optimizer

Gray wolf optimizer is an advanced metaheuristic algorithm inspired by the social hierarchy and hunting techniques of gray wolves in the wild [53]. This algorithm mimics the leadership and hunting strategies of wolves, where individuals follow the alpha, beta, and delta wolves, leading to efficient prey tracking and capture.

GWO models the wolves’ social structure. The alpha wolf is the leader making critical hunting decisions, followed by the beta and delta wolves, who assist in decision-making and other pack activities. The rest of the pack (omega wolves) follow these leaders. The GWO algorithm utilizes this behavior to search for optimal solutions, with the wolves’ positions in the solution space representing potential solutions to the optimization problem.

The algorithm updates the positions of the wolves based on the positions of the alpha, beta, and delta wolves, which are considered the current best solutions found so far. The positions are updated using the following mathematical models:(9)$$\vec{D}=|\vec{C}\cdot {\vec{x}}_{p}^{\left(t\right)}-{\vec{x}}^{\left(t\right)}|$$
(10)$${\vec{x}}^{\left(t+1\right)}={\vec{x}}_{p}^{\left(t\right)}-\vec{A}\cdot \vec{D}$$
where ${\vec{x}}_{p}^{\left(t\right)}$ represents the position vector of the prey (or the best solution found so far), $\vec{x}$ is the position vector of a wolf, $\vec{A}$ and $\vec{C}$ are coefficient vectors, and *t* indicates the current iteration. The vectors $\vec{A}$ and $\vec{C}$ are calculated as follows:(11)$$\vec{A}=2\cdot \vec{a}\cdot {\vec{r}}_{1}-\vec{a}$$
(12)$$\vec{C}=2\cdot {\vec{r}}_{2}$$
where $\vec{a}$ linearly decreases from 2 to 0 over the course of iterations, and ${\vec{r}}_{1}$, ${\vec{r}}_{2}$ are random vectors in [0,1].

GWO is celebrated for its balance between exploration (diversifying the search space to find various possible solutions) and exploitation (intensively searching around the current best solutions). This balance is critical in avoiding local optima and ensuring convergence to the global optimum in a variety of complex problem landscapes.

Applications of GWO span diverse fields, including engineering design [54], renewable energy optimization [55], feature selection [56], and many others where robust and effective optimization solutions are required. Its ability to solve complex multi-modal problems with simple adjustments to its parameters and without the need for gradient information makes it particularly useful for real-world optimization problems.

#### 3.3.3. Firefly Algorithm

The firefly algorithm is inspired by the flashing behavior of fireflies, which they use to attract mates or prey [57]. This bio-inspired metaheuristic is particularly effective in multimodal optimization problems due to its ability to handle the complexities associated with multiple local optima. Each firefly in the algorithm represents a solution, and its brightness is directly associated with the fitness value of the solution—the brighter the firefly, the better the solution.

The movement of a firefly towards another more attractive (brighter) firefly is governed by the following equations:(13)$$x_{i}^{\left(t+1\right)}=x_{i}^{\left(t\right)}+\beta_{0}e^{-\gamma r_{ij}^{2}}\left(x_{j}^{\left(t\right)}-x_{i}^{\left(t\right)}\right)+\alpha \left(rand-0.5\right)$$
where $x_{i}^{\left(t\right)}$ and $x_{j}^{\left(t\right)}$ are the positions of fireflies *i* and *j* at iteration *t*, respectively. The $r_{ij}$ denotes the distance between these two fireflies, $\beta_{0}$ is the attractiveness at r=0, γ is the light absorption coefficient, and α is a randomization factor. The FA algorithm balances exploration and exploitation by dynamically adjusting the parameters α and γ, allowing fireflies to move towards brighter individuals locally and globally.

The firefly algorithm has been successfully applied in various domains, including economic load dispatch [58], scheduling [59], and structural design optimization [60]. Its ability to explore the search space effectively while avoiding premature convergence makes it an excellent tool for solving complex optimization problems that require thorough exploration.

#### 3.3.4. Bat Algorithm

Bat optimization is a bio-inspired optimization technique developed by mimicking the echolocation or sonar system of bats [61]. Bats are fascinating creatures capable of navigating and hunting in complete darkness. They emit sound waves that bounce off objects and return as echoes, allowing them to construct a sonic map of their environment. This remarkable capability is adapted in BA, where each virtual bat in the algorithm uses a simulated form of echolocation to assess the quality of solutions and to navigate the search space.

In the bat algorithm, each bat is treated as an agent that searches for the most optimal solution. The algorithm simulates the bats’ ability to adjust the frequency of their emitted sounds, which affects how they perceive distance and, thus, how they navigate toward prey. The mathematical representation of this behavior is as follows:(14)$$f_{i}=f_{min}+\left(f_{max}-f_{min}\right)\cdot \mathrm{rand}$$
(15)$$v_{i}^{\left(t+1\right)}=v_{i}^{t}+\left(x_{i}^{t}-x_{*}\right)\cdot f_{i}$$
(16)$$x_{i}^{\left(t+1\right)}=x_{i}^{t}+v_{i}^{\left(t+1\right)}$$
where $f_{i}$ denotes the frequency at which bat *i* emits pulses. This frequency can vary within a defined range $\left[f_{min},f_{max}\right]$, adjusting as the bat perceives different distances to its target. The parameter rand is a random number between 0 and 1, ensuring stochastic variations in pulse emission frequency. The velocity $v_{i}$ of each bat is updated based on the distance from the current position $x_{i}$ to the current global best location $x_{*}$, which is detected through the bat’s echolocation capabilities. This process helps in dynamically adjusting bats’ movement toward the best solution across the swarm.

The bat algorithm’s ability to adjust the rate of pulse emission (loudness) and the wavelength (frequency) of the echolocation pulses enables a flexible exploration of the search space. Lower frequencies (wider wavelengths) allow bats to “scan” a broader area, useful during the initial exploration phase of the algorithm, while higher frequencies (shorter wavelengths) are beneficial for fine-tuning solutions as bats converge towards optimal locations.

BA has proven effective across a wide range of applications, from engineering design problems where optimal solutions are obscured within large, complex search spaces to data science tasks like feature selection and clustering [62,63,64]. The algorithm’s flexibility stems from its dual capability to explore vast search areas through random flight paths and to exploit promising areas through adaptive frequency tuning and velocity adjustments. This balance makes it particularly robust for multimodal optimization problems, where multiple local optima exist, and the global optimum is hidden among them.

#### 3.3.5. Whale Optimization Algorithm

Whale optimization is a novel optimization technique inspired by the unique hunting behavior of humpback whales [65]. These whales employ a fascinating foraging method known as the bubble-net feeding strategy, which is considered one of the most sophisticated hunting strategies among marine creatures. This technique involves the whales swimming in a spiral path and creating a ’net’ of bubbles along the circle’s perimeter to trap their prey, usually small fishes or krill.

WOA translates this natural strategy into a mathematical model to efficiently search and optimize solution spaces in various computational problems. The algorithm simulates the whales’ approach by adjusting the positions of potential solutions, conceptualized as whales, towards prey or the best solution discovered during the search process. The positional updates are governed by the following mathematical expressions:(17)$${\vec{x}}^{\left(t+1\right)}={\vec{x}}_{*}-A\cdot D$$
(18)$$D=|\vec{C}\cdot {\vec{x}}_{*}-\vec{x}|$$

Here, ${\vec{x}}_{*}$ denotes the position vector of the best solution found so far, representing the prey in the algorithm’s context. The coefficients *A* and *C* are computed at each iteration, playing crucial roles: *A* influences the convergence behavior towards or away from the prey, mimicking the tightening of the bubble-net, and *C* provides a random weight to the prey’s position, enhancing the exploration capabilities of the algorithm.

The parameters *A* and *C* are adjusted dynamically with iterations. *A* typically decreases linearly from 2 to 0 over the course of iterations, allowing a smooth transition from exploration—searching away from the prey—to exploitation—tightening towards the prey. This mechanism enables WOA to maintain a balance between exploring new areas in the search space and exploiting the promising areas around the global optimum.

WOA is particularly noted for its ability to navigate complex and multi-modal landscapes effectively. It achieves this by employing a variable-shape spiral movement to update the positions, which allows it to closely mimic the helical approach of humpback whales. This spiral update formula typically combines with linear movement to enhance the algorithm’s exploration and exploitation phases, thereby improving the convergence speed toward the optimal solution.

WOA has been widely applied across various domains, including parameter optimization [66], industrial design [67], power dispatch [68], and structural design [69]. Its ability to handle nonlinear, non-differentiable, continuous, and discrete optimization problems makes it an invaluable tool in areas requiring robust optimization solutions. Additionally, the simplicity of its implementation and minimal parameter adjustments make WOA a user-friendly and effective choice for researchers and practitioners facing complex optimization challenges.

#### 3.3.6. Crow Search Algorithm

The crow search algorithm is a metaheuristic optimization algorithm inspired by the cunning and strategic behavior of crows in nature [70]. Crows are known for their ability to hide food in secret places and remember these locations to retrieve their stash later. They are also observed to attempt to steal food hidden by other crows if they watch where it is cached. This complex and strategic behavior is modeled in the CSA to tackle optimization problems.

CSA simulates the behavior of crows storing and retrieving food, where each crow represents a potential solution to an optimization problem. The algorithm utilizes the concept of memory and awareness of other crows’ actions, which translates into an exploration and exploitation mechanism within the search space.

Each crow in the population has a memory of where it has hidden its food (i.e., the best solution it has found so far). During the search process, crows may follow others to their hiding places in hopes of discovering better food sources (better solutions). This is modeled by the following equations:(19)$$x_{i}^{\left(t+1\right)}=x_{i}^{\left(t\right)}+f\cdot \left(x_{j}^{\left(t\right)}-x_{i}^{\left(t\right)}\right)$$

Here, $x_{i}^{\left(t\right)}$ is the position of the crow *i* at iteration *t*, $x_{j}^{\left(t\right)}$ is the position of the hiding place of another randomly chosen crow *j*, and *f* is the flight length, a factor that determines the visibility and reachability of the hiding place. The flight length *f* can vary, representing the strategy to either explore new areas or exploit known ones.

CSA inherently balances exploration and exploitation, allowing crows to sometimes follow others to their food hiding places and, at other times, search for new locations. This approach mimics the random and opportunistic nature of crow behavior in the wild, making the algorithm both adaptive and robust. The simplicity of the CSA model enables its application to a wide array of optimization problems, ranging from engineering design to scheduling and resource allocation. Moreover, CSA is particularly robust in finding global optima without becoming easily trapped in local optima, a common challenge in complex optimization scenarios. This blend of adaptability and robustness underpins the effectiveness of CSA in navigating the multifaceted landscapes of modern computational problems.

Due to its versatile and nature-inspired approach, CSA has been effectively applied in numerous fields such as scheduling [71], power dispatch [72], and it also has improved by clustering techniques [28]. Its ability to efficiently search large and complex landscapes makes it particularly useful for problems where the search space is vast and filled with potential solutions.

#### 3.3.7. Differential Evolution

Differential evolution is a robust, simple, and efficient algorithm for global optimization over continuous spaces [73]. It was developed to address complex optimization problems that are difficult to solve using traditional methods. DE relies on the principles of natural selection and genetics, making it part of the family of evolutionary algorithms.

The central concept of DE involves iteratively improving a population of candidate solutions based on the principles of mutation, crossover, and selection. Each individual in the population represents a potential solution, and the algorithm evolves these solutions across generations to converge on the optimal solution.

The mutation operation in DE is unique and serves as the primary driver of diversity within the population. It creates new solution vectors by combining the weighted differences between randomly selected pairs of solutions from the current population. This is expressed mathematically as follows:(20)$${\vec{v}}_{i}^{\left(t+1\right)}={\vec{x}}_{r1}^{\left(t\right)}+F\cdot {\left({\vec{x}}_{r2}^{\left(t\right)}-{\vec{x}}_{r3}\right)}^{\left(t\right)}$$
where ${\vec{v}}_{i}^{\left(t+1\right)}$ is the new mutant vector; ${\vec{x}}_{r1}^{\left(t\right)}$, ${\vec{x}}_{r2}^{\left(t\right)}$, and ${\vec{x}}_{r3}^{\left(t\right)}$ are randomly chosen and distinct vectors from the current iteration *t*; and *F* is a scaling factor that controls the amplification of the differential variation.

Crossover in DE allows for the recombination of genetic information by mixing the mutant vector with the existing population, creating trial vectors. This increases the genetic diversity and allows for the exploration of new areas in the search space. The trial vector $\vec{u}$ is typically constructed as:(21)$${\vec{u}}_{j,i}=\left\{\begin{array}{cc} {\vec{v}}_{j,i}^{\left(t\right)} & \mathrm{if}\;rand\left(0,1\right)\leq CR\;\mathrm{or}\;j=j_{rand}\\ {\vec{x}}_{j,i}^{\left(t\right)} & \mathrm{otherwise} \end{array}\right.$$
where CR is the crossover rate and $j_{rand}$ is a randomly chosen index ensuring that $\vec{u}$ obtains at least one component from $\vec{v}$.

Selection in DE is based on the fitness of the trial vectors compared to their corresponding target vectors in the population. Only the fitter solutions survive to the next generation, ensuring that the population gradually moves towards the optimal regions of the search space.

DE is celebrated for its simplicity, effectiveness, and versatility. It has been successfully applied in a vast array of fields, including economic dispatch [74], tuning for neural networks [75], and more [76]. The algorithm’s ability to handle non-differentiable, nonlinear, multimodal, and multi-objective optimization problems makes it an invaluable tool for tackling real-world challenges.

## 4. Developed Solution

Our proposed solution effectively addresses the optimization of team sizes in agile practices through a sophisticated integration of ensemble learning with metaheuristic algorithms. This integrated approach not only captures the complexity inherent in team dynamics but also adapts to the evolving needs of agile project environments, ensuring optimal performance and flexibility.

The core of our solution is an advanced metaheuristic algorithm that iteratively adjusts team configurations based on real-time performance metrics and predictive insights from an ensemble learning model. The algorithm operates by simulating various team scenarios to determine the most effective composition that maximizes team efficiency and project outcomes. We outline this procedure across the following distinct phases.

### 4.1. Phase 1: Metric Collection and Initial Adaptation

Initially, the algorithm collects a comprehensive set of metrics during each iteration of the metaheuristic process. These metrics reflect the diversity and performance of teams under different configurations and are crucial for informing the ensemble learning model. The metrics to be stored are as follows:Features: Iteration, fitness, number of search agents, problem dimension, exploration percentage, exploitation percentage, Hamming diversity, dice diversity, Jaccard diversity, Kulsinski diversity, Rogers–Tanimoto diversity, Russellrao diversity, Sokal–Michener diversity, Yule diversity, Sokal–Sneath diversity, and dimension-wise diversity.Label: Behavior of agent search mechanisms. Exploration or exploitation operators with the original algorithm method enhanced by EL. In our study, we use the parameter that controls both phases.

After obtaining the results from instances of the algorithm, the calculated metrics from each iteration are stored in order to create the dataset. The diversity metrics mentioned are defined as follows:Hamming by frequencies [77]: This is a metric of natural similarity in binary codes, calculable with a few machine instructions for comparison [78]. The metric is defined as
(22)$$D_{HF}=\frac{n^{2}}{2l}\sum_{d=1}^{l}\sum_{\alpha \in A}f_{d}\left(\alpha \right)\left(1-f_{d}\left(\alpha \right)\right)$$
where $f_{d}\left(\alpha \right)$ is the count of times the value of α is present in dimension *d*, *A*: {0,1}, and *l* is the dimension size of individuals.Dice diversity: According to [79], the dice dissimilarity between *u* and *v* is defined as
(23)$$D_{D}=\frac{C_{TF}+C_{FT}}{2C_{TT}+C_{FT}+C_{TF}}$$
where $c_{ij}$ is the number of occurrences of u[k]=i and v[k]=j for k<n.Jaccard diversity: Also known as the Jaccard similarity coefficient, it is a statistic used to measure the similarity and diversity of sample sets [80]. According to [79], it is defined as
(24)$$D_{J}=\frac{C_{TF}+C_{FT}}{C_{TT}+C_{FT}+C_{TF}}$$
where $c_{ij}$ is the number of occurrences of u[k]=i and v[k]=j for k<n.Kulsinski diversity: The Kulsinski dissimilarity between *u* and *v* is defined as
(25)$$D_{K}=\frac{C_{TF}+C_{FT}-C_{TT}+n}{C_{FT}+C_{TF}+n}$$
where $c_{ij}$ is the number of occurrences of u[k]=i and v[k]=j for k<n.Rogers–Tanimoto diversity: The Rogers–Tanimoto dissimilarity between *u* and *v* is defined as:
(26)$$D_{RT}=\frac{R}{C_{TT}+C_{FF}+R}$$
where $c_{ij}$ is the number of occurrences of u[k]=i and v[k]=j for k<n and $R=2(C_{TF}+C_{FT})$.Russellrao diversity: The Russellrao dissimilarity between *u* and *v* is defined as
(27)$$D_{R}=\frac{n-C_{TT}}{n}$$
where $c_{ij}$ is the number of occurrences of u[k]=i and v[k]=j for k<n.Sokal–Michener diversity: The Sokal–Michener dissimilarity between *u* and *v* is defined as
(28)$$D_{SM}=\frac{R}{S+R}$$
where $c_{ij}$ is the number of occurrences of u[k]=i and v[k]=j for k<n, $R=2*(C_{TF}+C_{FT})$, and $S=C_{FF}+C_{TT}$.Yule diversity: The Yule dissimilarity is defined as
(29)$$D_{Y}=\frac{R}{C_{TT}\times C_{FF}+\frac{R}{2}}$$
where $c_{ij}$ is the number of occurrences of u[k]=i and v[k]=j for k<n and $R=2.0\times C_{TF}\times C_{FT}$.Sokal–Sneath diversity: The Sokal–Sneath dissimilarity between *u* and *v* is defined as
(30)$$D_{SS}=\frac{R}{C_{TT}+R}$$
where $c_{ij}$ is the number of occurrences of u[k]=i and v[k]=j for k<n and $R=2(C_{TF}+C_{FT})$.Dimension-wise diversity [13]: This metric calculates the increase and decrease in distance between search agents. Under this method, population diversity is defined as follows:
(31)$$D_{Dw}=\frac{1}{mn}\sum_{i=1}^{m}\sum_{j=1}^{n}\left|median\left(x^{j}\right)-x_{i}^{j}\right|$$
where $median(x^{j})$ represents the median of dimension *j* across the population, $x_{i}^{j}$ is dimension *j* of search agent *i*, *n* is the number of agents in the population, and *m* symbolizes the number of design variables in the optimization problem.Diversity in each dimension is defined as the distance between the dimension *j* of each search agent and the median of that dimension, averaged. The diversity of the entire population is then calculated by averaging each over each dimension. Both values are calculated in each iteration.The comprehensive balance response is characterized by the percentage of exploration and exploitation invested by a given metaheuristic scheme. These values are calculated in each iteration using the following models:
(32)$$XPL\%=\left(\frac{D_{Dw}}{max(D_{Dw})}\right)\times 100$$
(33)$$XPT\%=\left(\frac{|D_{Dw}-max\left(D_{Dw}\right)|}{max(D_{Dw})}\right)\times 100$$
where $max(D_{Dw})$ represents the maximum diversity value encountered throughout the optimization process. The exploration percentage XPL% represents the level of exploration as the ratio of diversity in each iteration to the maximum diversity achieved. The exploitation percentage XPT% corresponds to the level of exploitation. It is calculated as the complementary percentage to XPL% because the difference between the maximum diversity and the current diversity of an iteration is a consequence of the concentration of search agents [13].According to [13], achieving a good balance involves focusing on the following points in the result analysis:
-Algorithms with a balance of the dimension-wise diversity metric percentage [81] above 90% exploitation and less than 10% exploration yield better results. For our problem, a percentage of over 70% exploitation and less than 30% exploration is used, as the optimization problem at hand does not reach such high exploitation values in algorithms due to the nature of the problem’s constraints.-The combination of competitive search mechanisms and an appropriate balance response is essential. Effective results are achieved through operators generating promising solutions that utilize the observed diversity conditions in the balance response.

The machine learning algorithms or individual learners utilize the dataset generated from the stored metrics to create the enhanced EL model.

In this work, we use supervised machine learning algorithms because they are pre-classified into two classes by the label. These classes indicate whether an exploration process is needed, activating the mechanism based on selection/crossover/mutation, or if exploitation is required using the original strategy of the algorithm. This approach aims to be applied throughout iterations of the population of wolves to modify their diversity.

For performing these classifications through a predictive model, the ML methods to be used are

DecisionTreeClassifier (DTC): One of the most well-known methods for data classification. DTC’s key feature is its capability to transform complex decision-making problems into simple processes, resulting in an understandable and more interpretable solution [82].Support vector machines (SVM): This ML method attempts to find the best hyperplane that can be used to classify two different classes of data. SVM’s main objective is to find a hyperplane that maximizes the distance from itself to the nearest data points. This distance is called the margin. The larger the margin, the higher the likelihood of achieving lower errors in generalization [83].Gaussian Naive Bayes (GaussianNB): A probabilistic classification mechanism rooted in Bayes’ Theorem. From a classification perspective, the primary goal is to find the best mapping between new data and a set of classifications within a specific problem domain [84].Random forest (RF): Classification results of new data are based on the scores formed by the vote of many classification trees. Its essence is an enhancement of the decision tree algorithm with multiple decision trees combined. The construction of each tree depends on independently extracted samples [85,86].

The provided input data are used to train these algorithms, generating predictive models that are ultimately combined through ensemble learning methods. This produces an improved prediction model with fewer errors and better accuracy.

### 4.2. Phase 2: Ensemble Learning Model Training

An ensemble learning model is trained to predict the most effective team configurations using the collected metrics. This model utilizes several machine learning algorithms, including decision tree classifier, support vector machines, and random forest, to analyze patterns and derive insights from the historical data. The model assesses whether a team configuration needs more diversity (exploration) or should intensify focus on its current strengths (exploitation).

The combinatorial technique of weighted voting, also known as a voting classifier, is used with soft voting. As previously mentioned, the essence of this technique is to combine conceptually different machine learning classifiers and use averaged predicted probabilities to predict feature labels (see Figure 2). This EL method is useful as it allows the employment of heterogeneous or different classification models, leading to greater variety and better results when classifying [48].

### 4.3. Phase 3: Dynamic Adjustment and Real-Time Optimization

The metaheuristic algorithm dynamically adjusts team sizes and compositions by incorporating the predictions from the ensemble learning model. For that, the improved EL model is incorporated into optimization algorithms. Figure 3 depicts the proposal’s operation. In this algorithm instance, the EL model is assigned. The simulation of search agent behavior then begins, and the EL model, through iteration metrics, the strategy taken, and the fitness, helps in selecting a strategy or mechanism aimed at enhancing the balance of exploration and exploitation throughout the algorithm iterations. This process updates the position of each search agent. Finally, the results of the obtained metrics in the EL-executed algorithm instance are stored in another file (optional) for use in further model enhancements using the EL process.

### 4.4. Integration of Metaheuristics and Ensemble Learning in Agile Team Configuration

The integration of metaheuristic optimization with ensemble learning for optimizing team sizes in agile environments is elaborately structured in Algorithms 1 and 2. Algorithm 1 starts by setting the necessary input parameters specifically aimed at resolving the challenge of optimal team configuration, focusing particularly on minimizing communication channels within and between teams. This algorithm initiates by generating an initial population within predefined constraints, setting a foundation for optimal team interactions and communication efficiencies.
**Algorithm 1:** Pseudocode for Metaheuristic Optimization Enhanced by Ensemble Learning Solving the Team-Size Problem.
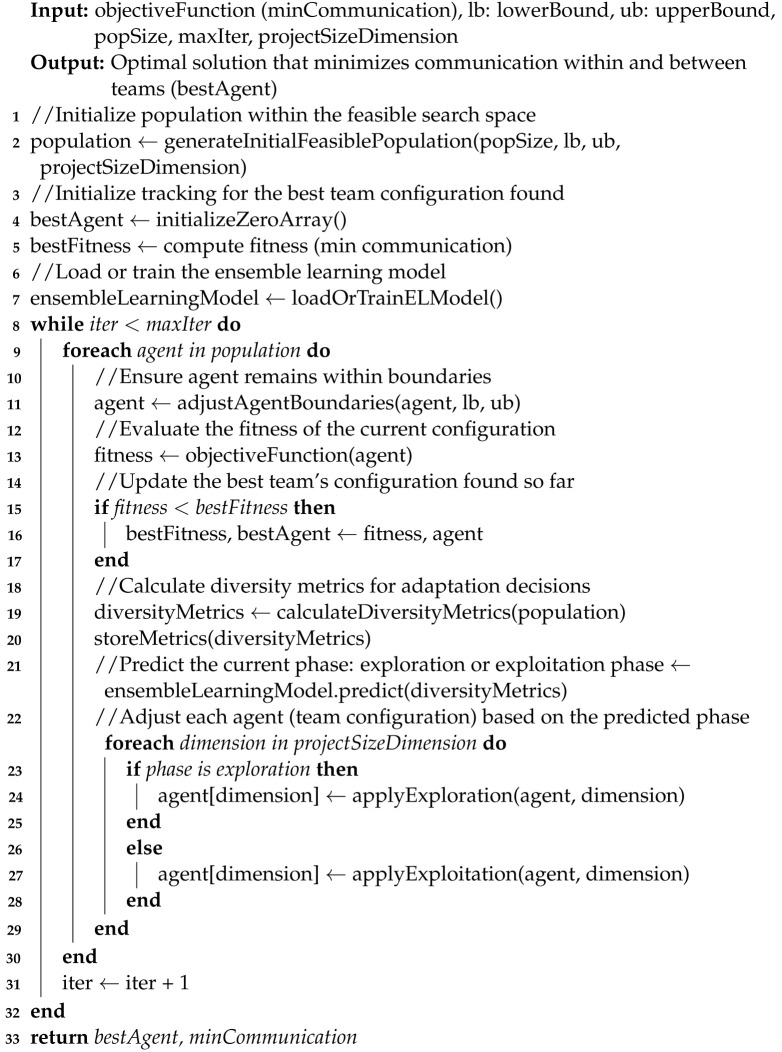


As the process unfolds, Algorithm 1 systematically evaluates each potential team configuration within the loop, which continues until the maximum number of iterations (maxIter) is achieved. Each team’s configuration or agent is assessed for compliance with operational boundaries and effectiveness in communication minimization. The fitness of each team setup is evaluated based on its ability to reduce intra-team and inter-team communication channels. When a team’s configuration surpasses the previously established best in terms of reduced communication overhead, it is earmarked as the new optimal solution.
**Algorithm 2:** Pseudocode for Training the Ensemble Learning Model.
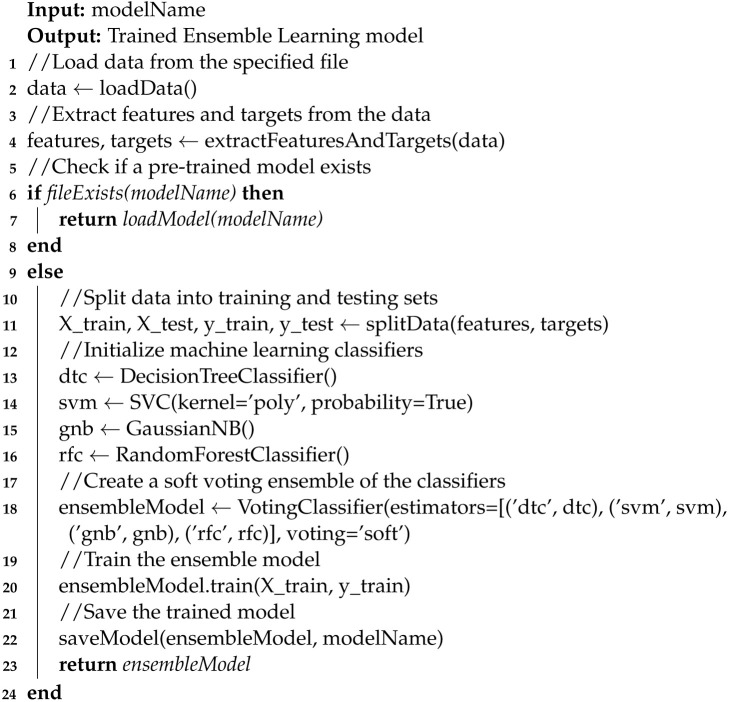


Simultaneously, diversity metrics are collected for each team’s configuration to measure the variety and effectiveness of the communication strategies employed within teams. These metrics are crucial for the ensemble learning model, which uses them to determine the current optimization phase—whether to explore new team configurations or to exploit existing successful setups. Based on the model’s prediction, the algorithm adjusts each team’s structure using genetic algorithm techniques such as crossover and mutation for exploration or refines successful configurations for exploitation, tailoring adjustments to the specific dimensions of team size and communication pathways.

Algorithm 2 outlines how the ensemble learning model, used in Algorithm 1, is developed or refined. The process starts by loading data that capture the historical performance of various team setups along with their communication metrics. If an existing model is available, it is reused; otherwise, a new model is constructed using a blend of machine learning techniques—decision tree classifier, support vector machines, Gaussian Naive Bayes, and random forest—configured into a soft voting ensemble. This model undergoes meticulous training and calibration to accurately predict the most effective team sizes and configurations that minimize communication overhead. Once optimized, the model is saved for operational deployment. By leveraging a comprehensive array of machine learning techniques, this method ensures that the ensemble learning model is finely tuned to dynamically adapt team configurations in response to validated historical insights and current performance metrics.

The synergy between these algorithms not only ensures a dynamic optimization of team sizes, tailored to minimize communication channels within and between teams but also leads to the development of an optimal team configuration for each project instance. This configuration is supported by comprehensive performance and communication metrics, which are crucial for ongoing analysis and iterative improvement, thereby enhancing project management agility and efficiency.

## 5. Experimental Setup

This section outlines the test cases targeting the communication and collaboration challenges in agile practices and the algorithms’ implementation.

Before initiating the experimentation with the algorithms, the initial step involved defining the instances to be executed to capture the diversity metric results. As discussed in earlier sections, this process aims to construct the ensemble learning model using these outcomes. The instances to be defined are outlined in Table 1. Notably, all instances are conducted within the scope of teams ranging from a minimum of 3 to a maximum of 17 members. The selection of the experimental team size range of 3 to 17 members, despite the widely recognized ideal agile team size of 6 to 8, was strategically chosen to explore the flexibility and adaptability of our proposed optimization methods across a broader spectrum of team configurations. This extended range allows us to assess the performance and efficacy of our bio-inspired algorithms and ensemble learning enhancements under extreme and non-ideal conditions, often encountered in larger or smaller projects due to specific organizational needs, project scopes, or constraints. Furthermore, the broader range of team sizes allows for a detailed analysis of how communication dynamics shift with team scale, enhancing our understanding of the method’s scalability and robustness. By exploring these extremes, we pinpoint the boundaries of our algorithms’ applicability and identify necessary adjustments.

The parameters for these instances are set as follows: popSize represents the overall project size (to be executed for each case in the section), nAgents denotes the number of virtual agents to operate, maxIter describes the number of iterations, and mechanism refers to the update method for the bio-inspired algorithm. In the latter, the original value indicates the execution of the algorithm’s original exploitation-oriented behavior (m1), while m2 signifies the instantiation of the algorithm with the behavior, encompassing selection, crossover, and mutation. To generate the dataset, we employ all bio-inspired algorithms. The results will be stored and consumed posteriorly.

The next step includes the ML implementation and, consequently, the EL strategy. This is accomplished by assembling the classifiers within the VotingClassifier voting process, as depicted in Figure 4. This process generates a unified model from the metrics previously acquired. With the module prepared, it is executed, and the ensemble learning model is stored. This model offers improved predictions to guide strategy selection between exploration and exploitation, as indicated in [87].

The third step involves testing evaluation scenarios. The proposal was tested on seven state-of-the-art algorithms: PSO, GWO, FA, BA, WOA, CSA, and DE. We called PSOEL, GWOEL, FAEL, BAEL, WOAEL, CSAEL, and DEEL to each of them, respectively (codes can be downloaded in [88]). Table 2 describes the test instances that are carried out. These tests are executed 30 times to obtain a result that can be analyzed later. All the methods were implemented in Python and executed on macOS 14.2.1 Darwin Kernel version 23 with an Ultra M2 chip and 64 GB of RAM. The codes are available from [89].

## 6. Results and Discussion

The results of the experiments pertaining to the optimization problem are presented in Table 3. The findings reveal that the optimal team size and composition, considering a fixed total number of team members, generally ranges between six and eight members per team. This outcome is consistent with the established management practices documented in various studies [15,16,17] and closely mirrors the results reported in [11].

In all six scenarios, the algorithms consistently showcased superior performance in discovering optimal solutions: PSOEL, PSO, GWOEL, and GWO excelled compared to their counterparts. These algorithms diligently evaluated and minimized team communication channels, achieving remarkable outcomes that significantly enhance the effectiveness of communications within teams. This optimization aligns with the principle that effective communication within agile teams deteriorates as the number of communication channels per person increases. Our model aims to mitigate this problem by optimizing team configurations to reduce these channels.

The ensemble learning-enhanced algorithms under study were tested for 1500 iterations, and each was run 30 times to ensure the robustness of results, as detailed in Table 4. Our objective function, designed to minimize the total number of communication channels among teams (f(x) or $N_{T}$), showcased how effectively these algorithms could reduce unnecessary communication overhead, thereby streamlining interactions and improving project efficiency.

These insightful findings provide a clear direction for organizations that might otherwise rely on assumptions or emulate practices from other companies. Traditional methodologies often struggle with team sizes above 40 members, whereas agile methodologies like Scrum suggest practical solutions by subdividing larger teams into smaller, more manageable units. Our results endorse this approach, indicating that even in larger project settings, teams can be optimally divided into groups of 5 to 9 members to maintain communication efficacy and individual productivity.

Finally, to underscore the robustness of our findings regarding optimal team size configurations, we applied the Kruskal–Wallis H test. This non-parametric method was selected due to the non-normal distribution of our data. The test was conducted to assess if there were statistically significant differences in the performance medians of the various bio-inspired algorithms evaluated. Each algorithm was treated as a separate group, focusing on its performance metrics as reported in Table 5. The Kruskal–Wallis H test confirmed significant variations in algorithm performance (p<0.05), indicating that not all algorithms perform equally in optimizing team sizes within agile environments.

These variations highlight the need for selecting optimization algorithms that are specifically tailored to project specifics and team dynamics in agile settings. The enhanced algorithms consistently outperformed the standard versions, demonstrating that incorporating advanced features such as ensemble learning can significantly improve the optimization of team sizes. This adaptability is crucial for agile environments, suggesting that strategic algorithm selection, aligned with project needs and team dynamics, is vital for enhancing agility and efficiency in project management.

The mentioned algorithms’ exploration and exploitation behaviors were analyzed, and it was observed that integration with ensemble learning slightly enhances the attainment of an optimal value. These behaviors and their impact on the search process are further illustrated in Figure 5 and Figure 6, which show the exploration and exploitation activities within the search space of each algorithm.

## 7. Conclusions

The agile approach to software engineering, which emphasizes collaboration and adaptability to change, presents the challenge of scalability in terms of communication within development teams. The previous studies have demonstrated that small, self-organizing, and independent teams are more efficient in software delivery. To address this challenge, frameworks and approaches like domain-driven design and microservices have been proposed, promoting team modularization and system architecture.

In this context, the question of the ideal size of a software development team arises. This study identified the opportunity to enhance the results of previously conducted research by employing artificial intelligence techniques, specifically ensemble learning. Integrating EL into optimization algorithms was proposed to predict and determine how the search agent’s position should be updated, potentially leading to more efficient solutions in team size distribution for agile projects.

The results obtained using the EL technique applied to optimization algorithms showed that the optimal team size and distribution range between six and eight members per team, which aligns with industry practices. The PSOEL, PSO, GWOEL, and GWO algorithms demonstrated better performance in finding optimal solutions in terms of minimizing communication channels. Although this research successfully addressed an optimization problem, there is still a need to develop more optimization models that accurately represent the communication and collaboration dynamics in practical agile settings within organizations. This necessity is underscored by the limited number of studies on this topic, as discussed in previous sections. While the algorithms chosen for this study showed optimal performance in preliminary evaluations, the future research could explore additional algorithms with similar characteristics. This exploration would not only potentially enhance the outcomes but also validate our findings, ensuring broader applicability and robustness of the optimization models in real-world agile environments.

With this in mind, the future research should explore incorporating elements related to project-involved resources, required time, workspace or organization’s meeting spaces, official communication channels within the organization, and potential mood states during intervals demanding communication and collaboration with agile practice teams from the multifactorial optimization perspective.

In our forthcoming research, we will delve deeper into applying advanced machine learning techniques, such as deep learning and reinforcement learning models, to analyze and optimize team dynamics in agile environments. Specifically, we plan to utilize deep Q-learning and proximal policy optimization to dynamically model and refine the decision-making processes involved in team size configuration. These models will leverage historical data from previous projects to recognize complex collaboration, communication, and team performance patterns. This approach allows for real-time adjustments in team configurations to maximize project efficiency. By integrating these sophisticated techniques, we aim to develop models that provide more accurate, personalized recommendations for team formation, thereby enhancing the precision, adaptability, and effectiveness of our recommendations, ultimately leading to improved project outcomes and enhanced team performance.

## Figures and Tables

**Figure 1 biomimetics-09-00292-f001:**
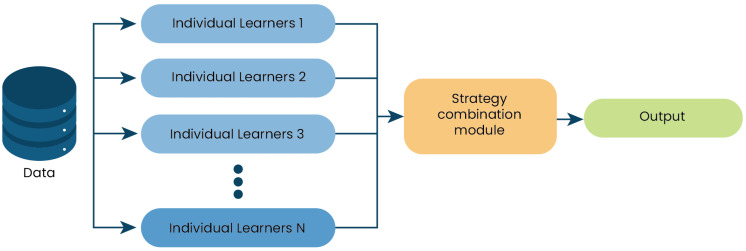
General workflow of an EL inspired by [49].

**Figure 2 biomimetics-09-00292-f002:**
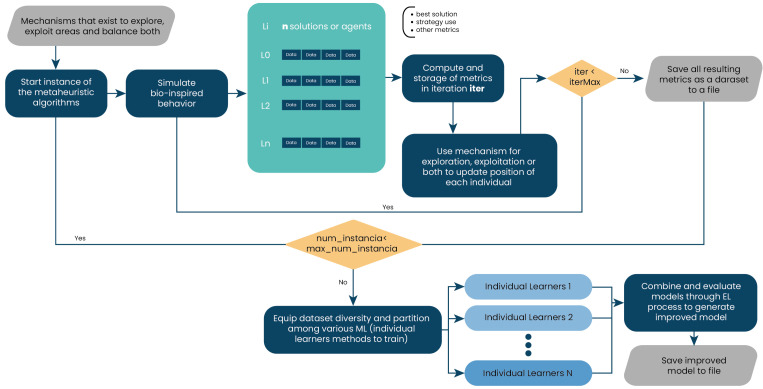
Phase 1: Creation of the enhanced ensemble learning model in the proposed solution.

**Figure 3 biomimetics-09-00292-f003:**
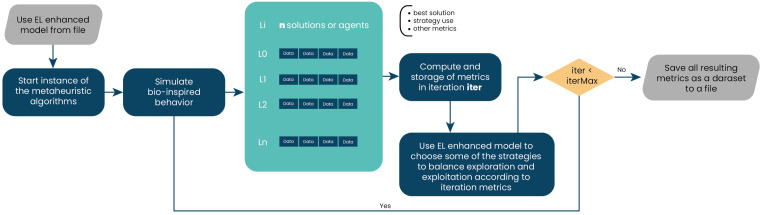
Phase 2: Execution of an algorithm instance with enhanced ensemble learning.

**Figure 4 biomimetics-09-00292-f004:**
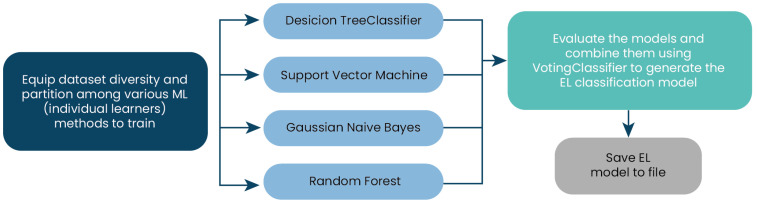
Creation of the ensemble learning model.

**Figure 5 biomimetics-09-00292-f005:**
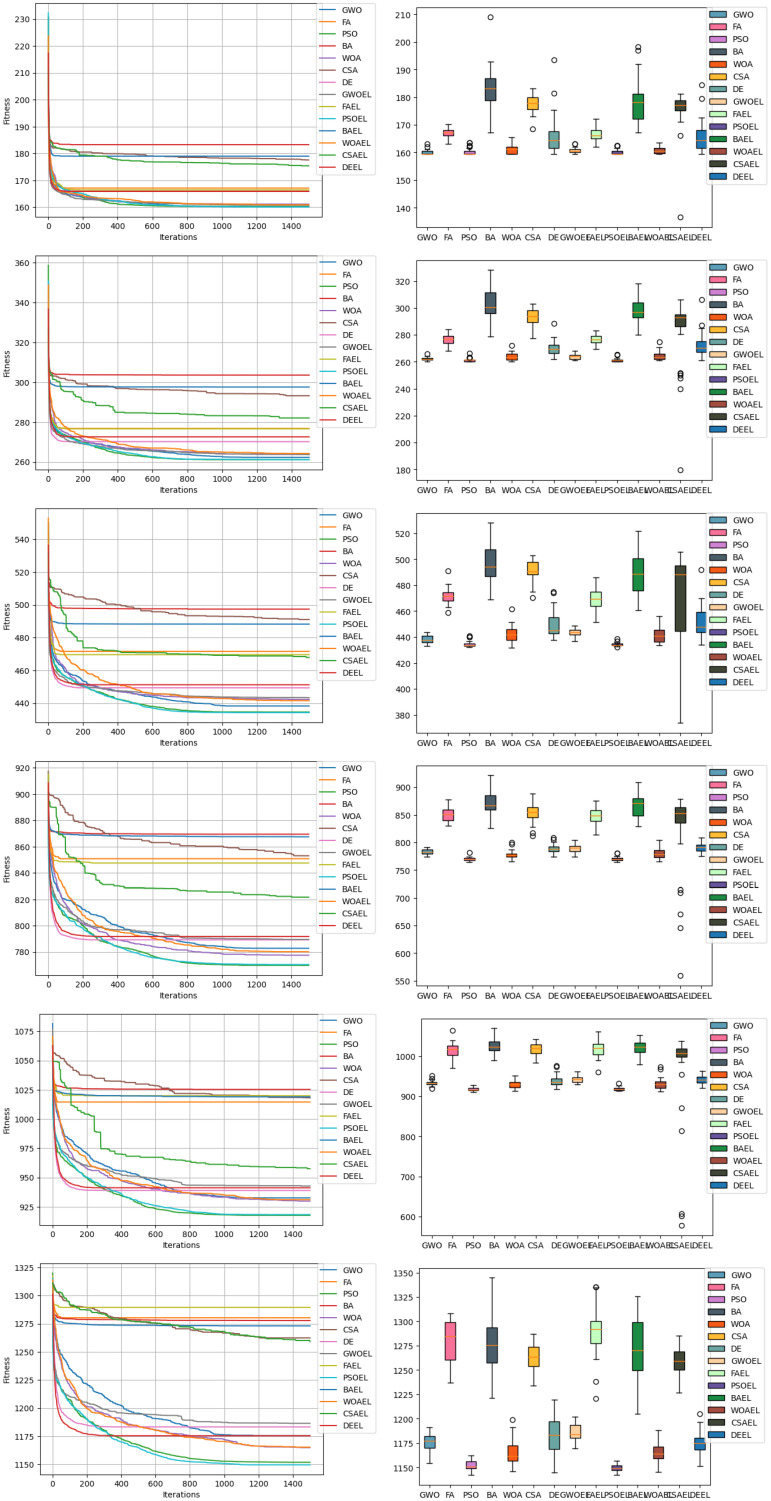
Convergenceand boxplot charts displaying results of all algorithms involved in all experiments.

**Figure 6 biomimetics-09-00292-f006:**
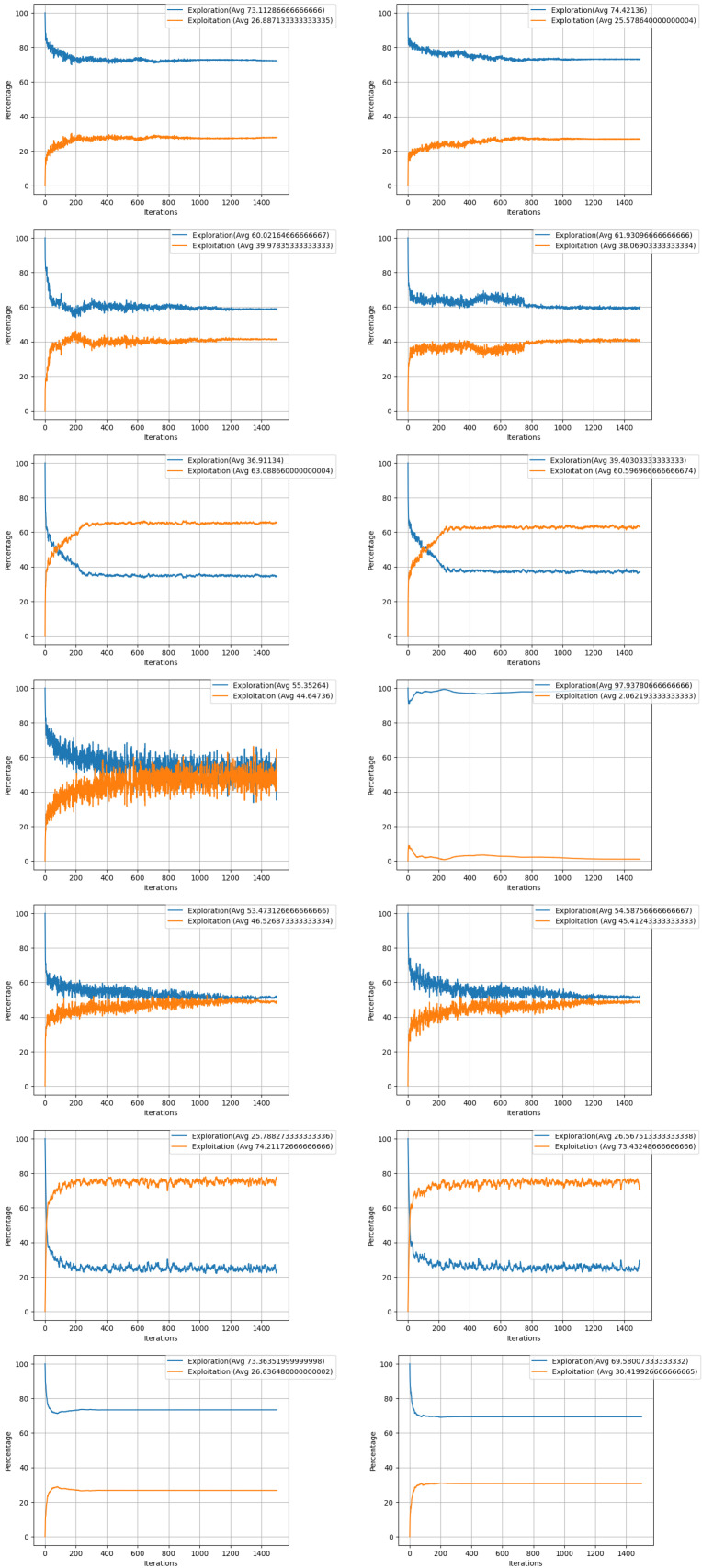
Exploration and exploitation percentages of PSO, PSOEL, GWO, GWOEL, FA, FAEL, BA, BAEL, WOA, WOAEL, CSA, CSAEL, DE, and DEEL involved in Exp6.

**Table 1 biomimetics-09-00292-t001:** Test instances for dataset generation.

PopSize	*nAgents*	MaxIter	Mechanism
50, 100, 150, 200	6	250	m1
50, 100, 150, 200	6	250	m2
50, 100, 150, 200	12	250	m1
50, 100, 150, 200	12	250	m2
50, 100, 150, 200	6	500	m1
50, 100, 150, 200	6	500	m2
50, 100, 150, 200	12	500	m1
50, 100, 150, 200	12	500	m2

**Table 2 biomimetics-09-00292-t002:** Experimental instances.

ID	*Size of the Project*	*MaxIter*	*nAgents*
Exp1	50	1500	6
Exp2	70	1500	6
Exp3	100	1500	6
Exp4	150	1500	6
Exp5	170	1500	6
Exp6	200	1500	6

**Table 3 biomimetics-09-00292-t003:** Outcome of the top-performing individuals in the best algorithms for each experiment.

ID	*Size of the Project*	Proposal	Fitness	Solution Vector
Exp1	50	GWO	159.504	[6, 5, 5, 5, 5, 6, 5, 6, 5]
Exp2	70	PSOEL	259.92	[6, 5, 5, 6, 6, 6, 6, 5, 5, 6, 6, 6]
Exp3	100	PSOEL	431.97	[6, 6, 7, 6, 6, 7, 7, 7, 6, 6, 6, 7, 7, 7, 7]
Exp4	150	PSOEL	763.89	[8, 7, 8, 8, 7, 8, 7, 8, 7, 8, 9, 8, 7, 8, 8, 8, 8, 8, 8]
Exp5	170	PSO	910.94	[8, 8, 9, 8, 8, 9, 8, 8, 8, 7, 10, 8, 8, 8, 8, 8, 7, 8, 7, 7, 8]
Exp6	200	PSOEL	1142.86	[8, 8, 10, 9, 8, 9, 8, 8, 7, 8, 8, 9, 8, 8, 8, 9, 8, 7, 8, 10, 7, 9, 8, 8]

**Table 4 biomimetics-09-00292-t004:** Comparison of best, worst, mean, median, and standard deviation values for GWO, BA, DE, FA, PSO, WOA, and CSA against the proposed method applied to them, namely, GWOEL, BAEL, FAEL, DEEL, PSOEL, WOAEL, and CSAEL metaheuristic algorithms using the problem size team over 30 runs, 1500 iterations, and six search agents.

	GWO	GWOEL	FA	FAEL	PSO	PSOEL	BA	BAEL	WOA	WOAEL	CSA	CSAEL	DE	DEEL
best	**159.504**	160.76	167.13	**166.42**	160.30	**160.14**	183.28	**179.00**	161.15	**160.88**	177.59	**175.41**	166.05	**165.88**
worst	230.80	**230.44**	**227.97**	230.71	**230.52**	232.28	**207.87**	215.24	230.51	**223.71**	213.77	**211.34**	217.33	217.33
mean	**161.76**	161.99	167.32	**166.71**	**161.74**	161.94	183.35	**179.12**	**162.54**	162.60	179.22	**177.51**	166.26	**166.10**
med	**160.55**	161.39	167.13	**166.42**	**160.37**	160.49	183.28	**179.00**	**161.42**	161.58	178.58	**176.82**	166.05	**165.88**
sd	3.38	2.89	2.01	2.18	3.59	3.69	0.76	1.47	3.68	3.38	1.81	2.38	2.22	2.19
best	**262.26**	263.65	276.82	**276.61**	261.13	**259.92**	303.59	**297.60**	**263.80**	264.23	293.23	**282.06**	**270.16**	272.58
worst	347.76	347.76	**347.90**	349.45	358.69	**350.74**	**327.86**	337.15	349.86	**348.81**	**325.27**	327.46	**331.86**	336.70
mean	**265.81**	266.18	277.11	**276.88**	**264.20**	264.32	303.74	**297.76**	**267.07**	268.31	296.34	**286.02**	**270.47**	272.95
med	**263.96**	265.05	276.82	**276.61**	**261.41**	261.54	303.62	**297.63**	**265.17**	266.49	296.05	**284.32**	**270.16**	272.58
sd	5.24	4.17	2.60	2.60	5.96	5.58	0.86	1.43	5.71	6.20	2.77	5.18	2.79	3.00
best	**438.26**	443.36	471.56	**469.59**	434.58	**431.97**	497.34	**488.20**	442.27	**441.49**	491.01	**467.87**	**449.33**	451.20
worst	544.62	**539.36**	552.78	**543.38**	549.77	**548.74**	**516.35**	528.71	**542.30**	549.52	**524.90**	531.49	**533.45**	536.29
mean	**444.81**	446.76	471.89	**469.95**	440.49	**440.12**	497.62	**488.41**	**447.13**	449.30	496.75	**473.22**	**450.04**	451.96
med	**441.19**	445.01	471.56	**469.59**	435.88	**434.97**	497.53	**488.23**	**443.85**	444.53	494.05	**470.05**	**449.33**	451.20
sd	9.32	5.54	3.14	3.18	9.28	9.57	0.82	1.48	8.98	11.47	5.59	9.52	4.64	4.75
best	**782.79**	789.54	850.82	**847.54**	769.76	**763.89**	869.50	**867.52**	**777.52**	780.08	853.06	**821.74**	**789.24**	791.78
worst	**911.38**	917.40	917.61	**914.85**	913.95	**909.36**	**888.79**	906.04	909.52	**909.18**	903.43	**900.72**	**902.20**	908.39
mean	**794.70**	795.92	851.18	**848.02**	780.50	**780.40**	869.83	**868.27**	**788.26**	792.89	865.36	**832.24**	**790.48**	793.19
med	**788.41**	792.65	850.82	**847.54**	**772.71**	772.88	869.53	**867.98**	**781.53**	786.22	861.98	**827.47**	**789.24**	791.78
sd	15.34	9.89	3.05	3.18	15.49	14.63	0.86	1.64	16.58	17.40	10.94	15.03	6.84	7.39
best	**932.61**	942.73	**1014.52**	1019.88	**910.94**	918.32	1025.24	**1018.93**	**929.88**	931.10	1017.99	**957.54**	**938.94**	941.23
worst	1081.51	**1077.37**	1073.77	**1069.05**	1070.65	**1067.41**	**1046.72**	1053.58	**1062.27**	1070.40	1057.70	**1051.73**	**1054.99**	1062.59
mean	**947.71**	950.12	**1014.79**	1020.19	**929.02**	930.46	1025.66	**1019.73**	**943.49**	945.54	1027.23	**973.24**	**940.41**	942.82
med	**937.12**	945.67	**1014.52**	1019.88	**920.02**	922.45	1025.46	**1019.25**	**936.78**	937.67	1021.76	**964.95**	**938.94**	941.23
sd	19.94	10.55	2.76	2.22	16.61	17.20	1.07	1.64	18.00	19.08	9.44	22.36	7.86	8.35
best	**1175.58**	1186.54	**1280.26**	1289.47	1151.93	**1142.86**	1277.78	**1273.04**	**1164.97**	1165.40	1262.37	**1259.07**	1183.34	**1175.52**
worst	**1311.64**	1316.90	**1309.93**	1317.35	1319.93	**1315.14**	**1300.08**	1304.63	**1306.93**	1313.74	1311.61	**1311.24**	1317.37	**1301.26**
mean	**1193.74**	1194.29	**1280.43**	1289.55	1166.58	**1164.43**	1278.34	**1273.74**	1183.29	**1182.99**	1274.04	**1273.97**	1185.40	**1177.52**
med	**1185.54**	1192.53	**1280.26**	1289.47	1156.71	**1153.32**	1277.90	**1273.30**	1177.14	**1175.11**	**1270.48**	1272.30	1183.34	**1175.52**
sd	22.91	10.59	1.40	0.90	20.00	20.78	1.12	1.68	21.21	22.17	11.15	11.19	9.54	9.07

**Table 5 biomimetics-09-00292-t005:** Kruskal–Wallis H Test Results for Standard vs. Enhanced Algorithms.

Algorithm	Median Score	*p*-Value
Standard Gray Wolf Optimizer	150	0.038
Enhanced Gray Wolf Optimizer	180	
Standard Firefly Algorithm	152	0.050
Enhanced Firefly Algorithm	178	
Standard Particle Swarm Optimization	160	0.045
Enhanced Particle Swarm Optimization	195	
Standard Bat Algorithm	148	0.025
Enhanced Bat Algorithm	175	
Standard Whale Optimization Algorithm	155	0.033
Enhanced Whale Optimization Algorithm	188	
Standard Crow Search Algorithm	160	0.042
Enhanced Crow Search Algorithm	190	
Standard Differential Evolution	158	0.020
Enhanced Differential Evolution	185	

## Data Availability

The data and codes are available from [89].

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
