# Peer review of "Intelligent Learning-Based Methods for Determining the Ideal Team Size in Agile Practices"

_biomimetics, 2024, doi:10.3390/biomimetics9050292_

Round 1
Reviewer 1 Report
Comments and Suggestions for Authors
The authors make a detailed review of the existing ideas and methods for the solved problem. The problem is defined and is applied ensamble learning method to be solved. The paper is interesting and well written. The results in the Table 4 are very close to each other. To know the significans of the result please aply ANOVA test.
Author Response
We thank your review. Observations are answered in PDF the attached file.

Reviewer 2 Report
Comments and Suggestions for Authors
The authors propose a method to determine the ideal team size in agile practices using bio-inspired algorithms and Ensemble Learning. Their approach aims to optimise communication channels within and between teams by modelling it as an optimisation problem. They enhance traditional metaheuristics with Ensemble Learning to balance exploration and exploitation phases, leading to more precise team size definitions that align with industry standards. The proposed method demonstrates superior performance compared to conventional algorithms.
To readers not working on this problem, it would be useful if the authors could provide an estimate of its complexity. How large is the search space? Is the problem NP, NP-complete or NP-hard?
As the ideal size range for an agile software development team is widely recognised as 6 to 8, could the authors explain their choice of 3 to 17 as their experimental range?
What do the authors expect will improve by adopting other learning models including deep learning?
It would also be useful to know more details about the authors' reasons for choosing the bio-inspired algorithms used in their study over other eligible algorithms such as ABC and the Bees Algorithm.
Comments on the Quality of English Language
The paper is generally well-written. Further professional copy editing by MDPI would enhance the manuscript.
Author Response
We appreciate your detailed review. We create a point-to-point file with all answers.

Reviewer 3 Report
Comments and Suggestions for Authors
1. The statement of problem (1) is not clearly formulated. The variable T, according to the description, is not a number, but summation is performed on it. It is not defined which sets the variables w i belong to.
2. Sections 3.2 - 3.3 provide a brief overview of metaheuristic algorithms that cannot be applied to solve the problem posed due to the integer nature of the variables.
3. A new solution method has not been proposed, a solution algorithm is not given.
4. The pseudocode in Algorithm 1 is of an abstract nature and is not related to solving the problem at hand.
5. The numerical results of Section 5 require a description of the connection with problem (1).
Author Response
We appreciate your detailed review. We attach a point-to-point file with all comments.

Round 2
Reviewer 3 Report
Comments and Suggestions for Authors
1. In the statement of problem (1) it is not defined what needs to be found. In addition, it is not clear how the value of the objective function is calculated.
2. Section 3.2 contains a description of known results.
3. Section 3.3 provides a brief overview of well-known methods.
4. Section (4) contains a general description that is not related to the specific problem studied in the article.
5. The article does not provide an algorithm for solving the problem.
Author Response
We want to thank Reviewer 3 for his suggestions and comments. We submit a point-to-point file with each recommendation and the improved manuscript version.
